# Interactive Control of Diverse Complex Characters with Neural Networks

**Igor Mordatch, Kendall Lowrey, Galen Andrew, Zoran Popovic, Emanuel Todorov**
Department of Computer Science, University of Washington
{mordatch,lowrey,galen,zoran,todorov}@cs.washington.edu

## Abstract

We present a method for training recurrent neural networks to act as near-optimal feedback controllers. It is able to generate stable and realistic behaviors for a range of dynamical systems and tasks – swimming, flying, biped and quadruped walking with different body morphologies. It does not require motion capture or task-specific features or state machines. The controller is a neural network, having a large number of feed-forward units that learn elaborate state-action mappings, and a small number of recurrent units that implement memory states beyond the physical system state. The action generated by the network is defined as velocity. Thus the network is not learning a control policy, but rather the dynamics under an implicit policy. Essential features of the method include interleaving supervised learning with trajectory optimization, injecting noise during training, training for unexpected changes in the task specification, and using the trajectory optimizer to obtain optimal feedback gains in addition to optimal actions.

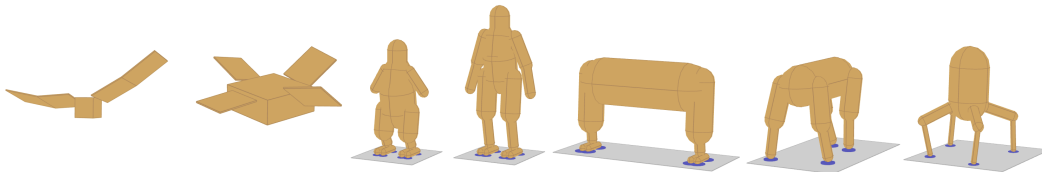

Figure 1: Illustration of the dynamical systems and tasks we have been able to control using the same method and architecture. See the video accompanying the submission.

## 1 Introduction

Interactive real-time controllers that are capable of generating complex, stable and realistic movements have many potential applications including robotic control, animation and gaming. They can also serve as computational models in biomechanics and neuroscience. Traditional methods for designing such controllers are time-consuming and largely manual, relying on motion capture datasets or task-specific state machines. Our goal is to automate this process, by developing universal synthesis methods applicable to arbitrary behaviors, body morphologies, online changes in task objectives, perturbations due to noise and modeling errors. This is also the ambitious goal of much work in Reinforcement Learning and stochastic optimal control, however the goal has rarely been achieved in continuous high-dimensional spaces involving complex dynamics.

Deep learning techniques on modern computers have produced remarkable results on a wide range of tasks, using methods that are not significantly different from what was used decades ago. The objective of the present paper is to design training methods that scale to larger and harder control problems, even if most of the components were already known. Specifically, we combine supervised

learning with trajectory optimization, namely Contact-Invariant Optimization (CIO) [12], which has given rise to some of the most elaborate motor behaviors synthesized automatically. Trajectory optimization however is an offline method, so the rationale here is to use a neural network to learn from the optimizer, and eventually generate similar behaviors online. There is closely related recent work along these lines [9, 11], but the method presented here solves substantially harder problems – in particular it yields stable and realistic locomotion in three-dimensional space, where previous work was applied to only two-dimensional characters. That this is possible is due to a number of technical improvements whose effects are analyzed below.

Control was historically among the earliest applications of neural networks, but the recent surge in performance has been in computer vision, speech recognition and other classification problems that arise in artificial intelligence and machine learning, where large datasets are available. In contrast, the data needed to learn neural network controllers is much harder to obtain, and in the case of imaginary characters and novel robots we have to synthesize the training data ourselves (via trajectory optimization). At the same time the learning task for the network is harder. This is because we need precise real-valued outputs as opposed to categorical outputs, and also because our network must operate not on i.i.d. samples, but in a closed loop, where errors can amplify over time and cause instabilities. This necessitates specialized training procedures where the dataset of trajectories and the network parameters are optimized together. Another challenge caused by limited datasets is the potential for over-fitting and poor generalization. Our solution is to inject different forms of noise during training. The scale of our problem requires cloud computing and a GPU implementation, and training that takes on the order of hours. Interestingly, we invest more computing resources in generating the data than in learning from it. Thus the heavy lifting is done by the trajectory optimizer, and yet the neural network complements it in a way that yields interactive real-time control.

Neural network controllers can also be trained with more traditional methods which do not involve trajectory optimization. This has been done in discrete action settings [10] as well as in continuous control settings [3, 6, 14]. A systematic comparison of these more direct methods with the present trajectory-optimization-based methods remains to be done. Nevertheless our impression is that networks trained with direct methods give rise to successful yet somewhat chaotic behaviors, while the present class of methods yield more realistic and purposeful behaviors.

Using physics based controllers allows for interaction, but these controllers need specially designed architectures for each range of tasks or characters. For example, for biped location common approaches include state machines and use of simplified models (such as the inverted pendulum) and concepts (such as zero moment or capture points) [21, 18]. For quadrupedal characters, a different set of state machines, contact schedules and simplified models is used [13]. For flying and swimming yet another set of control architectures, commonly making use of explicit cyclic encodings, have been used [8, 7]. It is our aim to unity these disparate approaches.

## 2   Overview

Let the state of the character be defined as $[\mathbf{q}\ \mathbf{f}\ \mathbf{r}]$, where $\mathbf{q}$ is the physical pose of the character (root position, orientation and joint angles), $\mathbf{f}$ are the contact forces being applied on the character by the ground, and $\mathbf{r}$ is the recurrent memory state of the character. The motion of the character is a state trajectory of length $T$ defined by $\mathbf{X} = \begin{bmatrix} \mathbf{q}^0\ \mathbf{f}^0\ \mathbf{r}^0\ ...\ \mathbf{q}^T\ \mathbf{f}^T\ \mathbf{r}^T \end{bmatrix}$. Let $\mathbf{X}^1, ..., \mathbf{X}^N$ be a collection of $N$ trajectories, each starting with different initial conditions and executing a different task (such as moving the character to a particular location).

We introduce a neural network control policy $\boldsymbol{\pi}_\theta : \mathbf{s} \mapsto \mathbf{a}$, parametrized by neural network weights $\theta$, that maps a sensory state of the character $\mathbf{s}$ at each point in time to an optimal action $\mathbf{a}$ that controls the character. In general, the sensory state can be designed by the user to include arbitrary informative features, but in this preliminary work we use the following simple and general-purpose representation:

$$\mathbf{s}^t = \begin{bmatrix} \mathbf{q}^t\ \mathbf{r}^t\ \dot{\mathbf{q}}^{t-1}\ \mathbf{f}^{t-1} \end{bmatrix} \qquad\qquad \mathbf{a}^t = \begin{bmatrix} \dot{\mathbf{q}}^t\ \dot{\mathbf{r}}^t\ \mathbf{f}^t \end{bmatrix},$$

where, e.g., $\dot{\mathbf{q}}^t \triangleq \mathbf{q}^{t+1} - \mathbf{q}^t$ denotes the instantaneous rate of change of $\mathbf{q}$ at time $t$. With this representation of the action, the policy directly commands the desired velocity of the character and applied contact forces, and determines the evolution of the recurrent state $\mathbf{r}$. Thus, our network learns both optimal controls and a model of dynamics simultaneously.

Let $C_i(\mathbf{X})$ be the total cost of the trajectory $\mathbf{X}$, which rewards accurate execution of task $i$ and physical realism of the character's motion. We want to jointly find a collection of optimal trajectories that each complete a particular task, along with a policy $\boldsymbol{\pi}_\theta$ that is able to reconstruct the sense and action pairs $\mathbf{s}^t(\mathbf{X})$ and $\mathbf{a}^t(\mathbf{X})$ of all trajectories at all timesteps:

$$\underset{\theta\,\mathbf{X}^1\,\dots\,\mathbf{X}^N}{\text{minimize}} \quad \sum_i C_i(\mathbf{X}^i) \qquad \text{subject to} \quad \forall\, i,t:\ \mathbf{a}^t(\mathbf{X}^i) = \boldsymbol{\pi}_\theta(\mathbf{s}^t(\mathbf{X}^i)). \tag{1}$$

The optimized policy parameters $\theta$ can then be used to execute policy in real-time and interactively control the character by the user.

## 2.1 Stochastic Policy and Sensory Inputs

Injecting noise has been shown to produce more robust movement strategies in graphics and optimal control [20, 6], reduce overfitting and prevent feature co-adaptation in neural network training [4], and stabilize recurrent behaviour of neural networks [5]. We inject noise in a principled way to aid in learning policies that do not diverge when rolled out at execution time.

In particular, we inject additive Gaussian noise into the sensory inputs $\mathbf{s}$ given to the neural network. Let the sensory noise be denoted $\varepsilon \sim \mathcal{N}(\mathbf{0}, \boldsymbol{\sigma}_\varepsilon^2 I)$, so the resulting noisy policy inputs are $\mathbf{s} + \varepsilon$. This is similar to denoising autoencoders [17] with one important difference: the change in input in our setting also induces a change in the optimal action to output. If the noise is small enough, the optimal action at nearby noisy states is given by the first order expansion

$$\mathbf{a}(\mathbf{s} + \varepsilon) = \mathbf{a} + \mathbf{a_s}\varepsilon, \tag{2}$$

where $\mathbf{a_s}$ (alternatively $\frac{d\mathbf{a}}{d\mathbf{s}}$) is the matrix of optimal feedback gains around $\mathbf{s}$. These gains can be calculated as a byproduct of trajectory optimization as described in section 3.2. Intuitively, such feedback helps the neural network trainer to learn a policy that can automatically correct for small deviations from the optimal trajectory and allows us to use much less training data.

## 2.2 Distributed Stochastic Optimization

The resulting constrained optimization problem (1) is nonconvex and too large to solve directly. We replace the hard equality constraint with a quadratic penalty with weight $\alpha$:

$$R(\mathbf{s}, \mathbf{a}, \theta, \varepsilon) = \frac{\alpha}{2} \left\| (\mathbf{a} + \mathbf{a_s}\varepsilon) - \boldsymbol{\pi}_\theta(\mathbf{s} + \varepsilon) \right\|^2, \tag{3}$$

leading to the relaxed, unconstrained objective

$$\underset{\theta\,\mathbf{X}^1\,\dots\,\mathbf{X}^N}{\text{minimize}} \quad \sum_i C_i(\mathbf{X}^i) + \sum_{i,t} R(\mathbf{s}^t(\mathbf{X}^i), \mathbf{a}^t(\mathbf{X}^i), \theta, \varepsilon^{i,t}). \tag{4}$$

We then proceed to solve the problem in block-alternating optimization fashion, optimizing for one set of variables while holding others fixed. In particular, we independently optimize for each $\mathbf{X}^i$ (trajectory optimization) and for $\theta$ (neural network regression).

As the target action $\mathbf{a} + \mathbf{a_s}\varepsilon$ depends on the optimal feedback gains $\mathbf{a_s}$, the noise $\varepsilon$ is resampled after optimizing each policy training sub-problem. In principle the noisy sensory state and corresponding action could be recomputed within the neural network training procedure, but we found it expedient to freeze the noise during NN optimization (so that the optimal feedback gains need not be passed to the NN training process). Similar to recent stochastic optimization approaches, we introduce quadratic proximal regularization terms (weighted by rate $\eta$) that keep the solution of the current iteration close to its previous optimal value. The resulting algorithm is

---
**Algorithm 1:** Distributed Stochastic Optimization

1  Sample sensor noise $\bar{\varepsilon}^{i,t}$ for each $t$ and $i$.
2  Optimize $N$ trajectories (sec 3): $\bar{\mathbf{X}}^i = \mathrm{argmin}_{\mathbf{X}}\, C_i(\mathbf{X}) + \sum_t R(\mathbf{s}^{i,t}, \mathbf{a}^{i,t}, \bar{\boldsymbol{\theta}}, \bar{\varepsilon}^{i,t}) + \frac{\eta}{2} \left\| \mathbf{X} - \bar{\mathbf{X}}^i \right\|^2$
3  Solve neural network regression (sec 4): $\bar{\boldsymbol{\theta}} = \mathrm{argmin}_{\boldsymbol{\theta}} \sum_{i,t} R(\bar{\mathbf{s}}^{i,t}, \bar{\mathbf{a}}^{i,t}, \boldsymbol{\theta}, \bar{\varepsilon}^{i,t}) + \frac{\eta}{2} \left\| \boldsymbol{\theta} - \bar{\boldsymbol{\theta}} \right\|^2$
4  Repeat.

---

Thus we have reduced a complex policy search problem in (1) to an alternating sequence of independent trajectory optimization and neural network regression problems, each of which are well-studied and allow the use of existing implementations. While previous work [9, 11] used ADMM or dual gradient descent to solve similar optimization problems, it is non-trivial to adapt them to asynchronous and stochastic setting we have. Despite potentially slower rate, we still observe convergence as shown in section 8.1.

## 3 Trajectory Optimization

We wish to find trajectories that start with particular initial conditions and execute the task, while satisfying physical realism of the character's motion. The existing approach we use is Contact-Invariant Optimization (CIO) [12], which is a direct trajectory optimization method based on inverse dynamics. Define the total cost for a trajectory $\mathbf{X}$:

$$C(\mathbf{X}) = \sum_t c(\phi^t(\mathbf{X})), \tag{5}$$

where $\phi^t(\mathbf{X})$ is a function that extracts a vector of features (such as root forces, contact distances, control torques, etc.) from the trajectory at time $t$ and $c(\phi)$ is the state cost over these features.

Physical realism is achieved by satisfying equations of motion, non-penetration, and force complementarity conditions at every point in the trajectory [12]:

$$H(\mathbf{q})\ddot{\mathbf{q}} + C(\mathbf{q}, \dot{\mathbf{q}}) = \boldsymbol{\tau} + J^\top(\mathbf{q}, \dot{\mathbf{q}})\mathbf{f}, \qquad \mathbf{d}(\mathbf{q}) \geq \mathbf{0}, \qquad \mathbf{d}(\mathbf{q})^\top \mathbf{f} = \mathbf{0}, \qquad \mathbf{f} \in \mathbf{K}(\mathbf{q}) \tag{6}$$

where $\mathbf{d}(\mathbf{q})$ is the distance of the contact to the ground and $\mathbf{K}$ is the contact friction cone. These constraints are implemented as soft constraints, as in [12] and are included in $C(\mathbf{X})$. Initial conditions are also implemented as soft constraints in $C(\mathbf{X})$. Additionally we want to make sure the task is satisfied, such as moving to a particular location while minimizing effort. These task costs are the same for all our experiments and are described in section 8. Importantly, CIO is able to find solutions with trivial initializations, which makes it possible to have a broad range of characters and behaviors without requiring hand-designed controllers or motion capture for initialization.

### 3.1 Optimal Trajectory

The trajectory optimization problem consists of finding the optimal trajectory parameters $\mathbf{X}$ that minimize the total cost (5) with objective (3) now folded into $C$ for simplicity:

$$\mathbf{X}^* = \underset{\mathbf{X}}{\operatorname{argmin}} C(\mathbf{X}). \tag{7}$$

We solve the above optimization problem using Newton's method, which requires the gradient and Hessian of the total cost function. Using the chain rule, these quantities are

$$C_{\mathbf{X}} = \sum_t c_\phi^t \phi_{\mathbf{X}}^t \qquad C_{\mathbf{X}\mathbf{X}} = \sum_t (\phi_{\mathbf{X}}^t)^\top c_{\phi\phi}^t \phi_{\mathbf{X}}^t + c_\phi^t \phi_{\mathbf{X}\mathbf{X}}^t \approx \sum_t (\phi_{\mathbf{X}}^t)^\top c_{\phi\phi}^t \phi_{\mathbf{X}}^t$$

where the truncation of the last term in $C_{\mathbf{X}\mathbf{X}}$ is the common Gauss-Newton Hessian approximation [1]. We choose cost functions for which $c_\phi$ and $c_{\phi\phi}$ can be calculated analytically. On the other hand, $\phi_{\mathbf{X}}$ is calculated by finite differencing. The optimum can then be found by the following recursion:

$$\mathbf{X}^* = \mathbf{X}^* - C_{\mathbf{X}\mathbf{X}}^{-1} C_{\mathbf{X}}. \tag{8}$$

Because this optimization is only a sub-problem (step 2 in algorithm 1), we don't run it to convergence, and instead take between one and ten iterations.

### 3.2 Optimal Feedback Gains

In addition to the optimal trajectory, we also need to find optimal feedback gains $\mathbf{a_s}$ necessary to generate optimal actions for noisy inputs in (2). While these feedback gains are a byproduct of indirect trajectory optimization methods such as LQG, they are not an obvious result of direct trajectory optimization methods like CIO. While we can use Linear Quadratic Gaussian (LQG)

pass around our optimal solution to compute these gains, this is inefficient as it does not make use of computation already performed during direct trajectory optimization. Moreover, we found the resulting process can produce very large and ill-conditioned feedback gains. One could change the objective function for the LQG pass when calculating feedback gains to make them smoother (for example, by adding explicit trajectory smoothness cost), but then the optimal actions would be using feedback gains from a different objective. Instead, we describe a perturbation method that reuses computation done during direct trajectory optimization, also producing better-conditioned gains. This is a general method for producing feedback gains that stabilize resulting optimal trajectories and can be useful for other applications.

Suppose we perturb a certain aspect of optimal trajectory $\mathbf{X}$, such that the sensory state changes: $\mathbf{s}(\mathbf{X}) = \bar{\mathbf{s}}$. We wish to find how the optimal action $\mathbf{a}(\mathbf{X})$ will change given this perturbation. We can enforce the perturbation with a soft constraint of weight $\lambda$, resulting in an augmented total cost:

$$\tilde{C}(\mathbf{X}, \bar{\mathbf{s}}) = C(\mathbf{X}) + \frac{\lambda}{2} \left\| \mathbf{s}(\mathbf{X}) - \bar{\mathbf{s}} \right\|^2. \tag{9}$$

Let $\tilde{\mathbf{X}}(\bar{\mathbf{s}}) = \operatorname{argmin}_{\mathbf{X}}^* \tilde{C}(\mathbf{X}^*)$ be the optimum of the augmented total cost. For $\bar{\mathbf{s}}$ near $\mathbf{s}(\mathbf{X})$ (as is the case with local feedback control), the minimizer of augmented cost is the minimizer of a quadratic around optimal trajectory $\mathbf{X}$

$$\tilde{\mathbf{X}}(\bar{\mathbf{s}}) = \mathbf{X} - \tilde{C}_{\mathbf{XX}}^{-1}(\mathbf{X}, \bar{\mathbf{s}})\tilde{C}_{\mathbf{X}}(\mathbf{X}, \bar{\mathbf{s}}) = \mathbf{X} - (C_{\mathbf{XX}} + \lambda \mathbf{s}_{\mathbf{X}}^\top \mathbf{s}_{\mathbf{X}})^{-1}(C_{\mathbf{X}} + \lambda \mathbf{s}_{\mathbf{X}}^\top (\mathbf{s}(\mathbf{X}) - \bar{\mathbf{s}})),$$

where all derivatives are calculated around $\mathbf{X}$. Differentiating the above w.r.t. $\bar{\mathbf{s}}$,

$$\tilde{\mathbf{X}}_{\bar{\mathbf{s}}} = \lambda(C_{\mathbf{XX}} + \lambda \mathbf{s}_{\mathbf{X}}^\top \mathbf{s}_{\mathbf{X}})^{-1}\mathbf{s}_{\mathbf{X}}^\top = C_{\mathbf{XX}}^{-1}\mathbf{s}_{\mathbf{X}}^\top (\mathbf{s}_{\mathbf{X}} C_{\mathbf{XX}}^{-1} \mathbf{s}_{\mathbf{X}}^\top + \frac{1}{\lambda}I)^{-1},$$

where the last equality follows from Woodbury identity and has the benefit of reusing $C_{\mathbf{XX}}^{-1}$, which is already computed as part of trajectory optimization. The optimal feedback gains for $\mathbf{a}$ are $\mathbf{a}_{\bar{\mathbf{s}}} = \mathbf{a}_{\mathbf{X}}\tilde{\mathbf{X}}_{\bar{\mathbf{s}}}$. Note that $\mathbf{s}_{\mathbf{X}}$ and $\mathbf{a}_{\mathbf{X}}$ are subsets of $\phi_{\mathbf{X}}$, and are already calculated as part of trajectory optimization. Thus, computing optimal feedback gains comes at very little additional cost.

Our approach produces softer feedback gains according to parameter $\lambda$ without modifying the cost function. The intuition is that instead of holding perturbed initial state fixed (as LQG does, for example), we make matching the initial state a soft constraint. By weakening this constraint, we can modify initial state to better achieve the master cost function without using very aggressive feedback.

## 4  Neural Network Policy Regression

After performing trajectory optimization, we perform standard regression to fit a neural network to the noisy fixed input and output pairs $\{\mathbf{s} + \boldsymbol{\varepsilon},\ \mathbf{a} + \mathbf{a_s}\boldsymbol{\varepsilon}\}^{i,t}$ for each timestep and trajectory. Our neural network policy has a total of $K$ layers, hidden layer activation function $\sigma$ (*tanh*, in the present work) and hidden units $\mathbf{h}^k$ for layer $k$. To learn a model that is robust to small changes in neural state, we add independent Gaussian noise $\boldsymbol{\gamma}^k \sim \mathcal{N}(\mathbf{0}, \boldsymbol{\sigma}_\gamma^2 I)$ with variance $\boldsymbol{\sigma}_\gamma^2$ to the neural activations at each layer during training. Wager et al. [19] observed this noise model makes hidden units tend toward saturated regions and less sensitive to precise values of individual units.

As with the trajectory optimization sub-problems, we do not run the neural network trainer to convergence but rather perform only a single pass of batched stochastic gradient descent over the dataset before updating the parameters $\boldsymbol{\theta}$ in step 3 of Algorithm 1.

All our experiments use 3 hidden layer neural networks with 250 hidden units in each layer (other network sizes are evaluated in section 8.1). The neural network weight matrices are initialized with a spectral radius of just above 1, similar to [15, 5]. This helps to make sure initial network dynamics are stable and do not vanish or explode.

## 5  Training Trajectory Generation

To train a neural network for interactive use, we required a data set that includes dynamically changing task's goal state. The task, in this case, is the locomotion of a character to a movable goal

position controlled by the user. (Our character's goal position was always set to be the origin, which encodes the characters state position in the goal position's coordinate frame. Thus the "origin" may shift relative to the character, but this keeps behavior invariant to the global frame of reference.)

Our trajectory generation creates a dataset consisting of trials and segments. Each trial $k$ starts with a reference physical pose and null recurrent memory $[\mathbf{q}\ \dot{\mathbf{q}}\ \mathbf{r}]^{\text{init}}$ and must reach goal location $\mathbf{g}^{k,0}$. After generating an optimal trajectory $\mathbf{X}^{k,0}$ according to section 3, a random timestep $t$ is chosen to branch a new segment with $[\mathbf{q}\ \dot{\mathbf{q}}\ \mathbf{r}]^{t}$ used as the initial state. A new goal location $\mathbf{g}^{k,1}$ is also chosen randomly for optimal trajectory $\mathbf{X}^{k,1}$.

This process represents the character changing direction at some point along its original trajectory plan: "interaction" in this case is simply a new change in goal position. This technique allows for our initial states and goals to come from the distribution that reflects the character's typical motion. In all our experiments, we use between 100 to 200 trials, each with 5 branched segments.

# 6 Distributed Training Architecture

Our training algorithm was implemented in a asynchronous, distributed architecture, utilizing a GPU for neural network training. Simple parallelism was achieved by distributing the trajectory optimization processes to multiple node machines, while the resulting data was used to train the NN policy on a single GPU node.

Amazon Web Service's EC2 3.8xlarge instances provided the nodes for optimization, while a g2.2xlarge instance provided the GPU. Utilizing a star-topology with the GPU instance at the center, a Network File System server distributes the training data $\mathbf{X}$ and network parameters $\theta$ to necessary processes within the cluster. Each optimization node is assigned a subset of the total trials and segments for the given task. This simple usage of files for data storage meant no supporting infrastructure other than standard file locking for concurrency.

We used a custom GPU implementation of stochastic gradient descent (SGD) to train the neural network control policy. For the first training epoch, all trajectories and action sequences are loaded onto the GPU, randomly shuffling the order of the frames. Then the neural network parameters $\theta$ are updated using batched SGD in a single pass over the data to reduce the objective in (4). At the start of subsequent training epochs, trajectories which have been updated by one of the trajectory optimization processes (and injected with new sensor noise $\varepsilon$) are reloaded.

Although this architecture is asynchronous, the proximal regularization terms in the objective prevent the training data and policy results from changing too quickly and keep the optimization from diverging. As a result, we can increase our training performance linearly for the size of cluster we are using, to about 30 optimization nodes per GPU machine. We run the overall optimization process until the average of 200 trajectory optimization iterations has been reached across all machines. This usually results in about 10000 neural network training epochs, and takes about 2.5 hours to complete, depending on task parameters and number of nodes.

# 7 Policy Execution

Once we find the optimal policy parameters $\boldsymbol{\theta}$ offline, we can execute the resulting policy in real-time under user control. Unlike non-parametric methods like motion graphs or Gaussian Processes, we do not need to keep any trajectory data at execution time. Starting with an initial state $\mathbf{x}^0$, we compute sensory state $\mathbf{s}$ and query the policy (without noise) for the desired action $\begin{bmatrix} \dot{\mathbf{q}}^{\text{des}}\ \dot{\mathbf{r}}^{\text{des}}\ \mathbf{f} \end{bmatrix}$.

To evolve the physical state of the system, we directly optimize the next state $\mathbf{x}^1$ to match $\dot{\mathbf{q}}^{\text{des}}$ while satisfying equations of motion

$$\mathbf{x}^1 = \operatorname*{argmin}_{\mathbf{x}} \left\|\dot{\mathbf{q}} - \dot{\mathbf{q}}^{\text{des}}\right\|^2 + \left\|\dot{\mathbf{r}} - \dot{\mathbf{r}}^{\text{des}}\right\|^2 + \left\|\mathbf{f} - \mathbf{f}^{\text{des}}\right\|^2 \quad \text{subject to (6)}$$

Note that this is simply the optimization problem (7) with horizon $T = 1$, which can be solved at real-time rates and does not require any additional implementation. This approach is reminiscent of feature-based control in computer graphics and robotics.

Because our physical state evolution is a result of optimization (similar to an implicit integrator), it does not suffer from instabilities or divergence as Euler integration would, and allows the use of larger timesteps (we use $\Delta t$ of 50ms in all our experiments). In the current work, the dynamics constraints are enforced softly and thus may include some root forces in simulation.

# 8    Results

This algorithm was applied to learn a policy that allows interactive locomotion for a range of very different three-dimensional characters. We used a single network architecture and parameters to create all controllers without any specialized initializations. While the task is locomotion, different character types exhibit very different behaviors. The experiments include three-dimensional swimming and flying characters as well as biped and quadruped walking tasks. Unlike in two-dimensional scenarios, it is much easier for characters to fall or go into unstable regions, yet our method manages to learn successful controllers. We strongly suggest viewing the supplementary video for examples of resulting behaviors.

The swimming creature featured four fins with two degrees of freedom each. It is propelled by lift and drag forces for simulated water density of 1000kg/m$^3$. To move, orient, or maintain position, controller learned to sweep down opposite fins in a cyclical patter, as in treading water. The bird creature was a modification of the swimmer, with opposing two-segment wings and the medium density changed changed to that of air (1.2kg/m$^3$). The learned behavior that emerged is cyclical flapping motion (more vigorous now, because of the lower medium density) as well as utilization of lift forces to coast to distant goal positions and modulation of flapping speed to change altitude.

Three bipedal creatures were created to explore the controller's function with respect to contact forces. Two creatures were akin to a humanoid - one large and one small, both with arms - while the other had a very wide torso compared to its height. All characters learned to walk to the target location and orientation with a regular, cyclic gait. The same algorithm also learned a stereotypical trot gait for a dog-like and spider-like quadrupeds. This alternating left/right footstep cyclic behavior for bipeds or trot gaits for quadrupeds emerged without any user input or hand-crafting.

The costs in the trajectory optimization were to reach goal position and orientation while minimizing torque usage and contact force magnitudes. We used the MuJoCo physics simulator [16] engine for our dynamics calculations. The values of the algorithmic constants used in all experiments are $\sigma_\varepsilon = 10^{-2}$ $\sigma_\gamma = 10^{-2}$ $\alpha = 10$ $\lambda = 10^2$ $\eta = 10^{-2}$.

## 8.1    Comparative Evaluation

We show the performance of our method on a biped walking task in figure 2 under **full method** case. To test the contribution of our proposed joint optimization technique, we compared our algorithm to naive neural network training on a static optimal trajectory dataset. We disabled the neural network and generated optimal trajectories as according to 5. Then, we performed our regression on this static data set with no trajectories being re-optimized. The results are shown in **no joint** case. We see that at test time, our full method performs two orders of magnitude better than static training. To test the contribution of noise injection, we used our full method, but disabled sensory and hidden unit noise (sections 2.1 and 4). The results are under **no noise** case. We observe typical overfitting, with good training performance, but very poor test performance. In practice, both ablations above lead to policy rollouts that quickly diverge from expected behavior.

Additionally, we have compared the performance of different policy network architectures on the biped walking task by varying the number of layers and hidden units. The results are shown in table 1. We see that 3 hidden layers of 250 units gives the best performance/complexity tradeoff.

Model-predictive control (MPC) is another potential choice of a real-time controller for task-driven character behavior. In fact, the trajectory costs for both MPC and our method are very similar. The resulting trajectories, however, end up being different: MPC creates effective trajectories that are not cyclical (both are shown in figure 3 for a bird character). This suggests a significant nullspace of task solutions, but from all these solutions, our joint optimization - through the cost terms of matching the neural network output - act to regularize trajectory optimization to predictable and less chaotic behaviors.

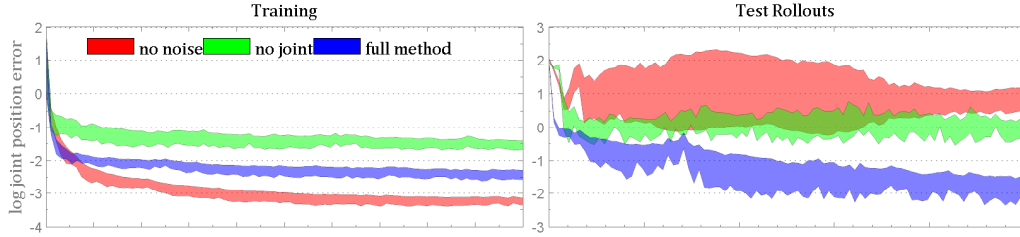

Figure 2: Performance of our full method and two ablated configurations as training progresses over 10000 neutral network updates. Mean and variance of the error is over 1000 training and test trials.

| 10 neurons | $0.337 \pm 0.06$ |
|---|---|
| 25 neurons | $0.309 \pm 0.06$ |
| 100 neurons | $0.186 \pm 0.02$ |
| 250 neurons | $0.153 \pm 0.02$ |
| 500 neurons | $0.148 \pm 0.02$ |

| 1 layer | $0.307 \pm 0.06$ |
|---|---|
| 2 layers | $0.253 \pm 0.06$ |
| 3 layers | $0.153 \pm 0.02$ |
| 4 layers | $0.158 \pm 0.02$ |

(a) Increasing Neurons per layer with 4 layers

(b) Increasing Layers with 250 neurons per layer

Table 1: Mean and variance of joint position error on test rollouts with our method after training with different neural network configurations.

# 9    Conclusions and Future Work

We have presented an automatic way of generating neural network parameters that represent a control policy for physically consistent interactive character control, only requiring a dynamical character model and task description. Using both trajectory optimization and stochastic neural networks together combines correct behavior with real-time interactive use. Furthermore, the same algorithm and controller architecture can provide interactive control for multiple creature morphologies.

While the behavior of the characters reflected efficient task completion in this work, additional modifications could be made to affect the style of behavior – costs during trajectory optimization can affect how a task is completed. Incorporation of muscle actuation effects into our character models may result in more biomechanically plausible actions for that (biologically based) character. In addition to changing the character's physical characteristics, we could explore different neural network architectures and how they compare to biological systems. With this work, we have networks that enable diverse physical action, which could be augmented to further reflect biological sensorimotor systems. This model could be used to experiment with the effects of sensor delays and the resulting motions, for example [2].

This work focused on locomotion of different creatures with the same algorithm. Previous work has demonstrated behaviors such as getting up, climbing, and reaching with the same trajectory optimization method [12]. Real-time policies using this algorithm could allow interactive use of these behaviors as well. Extending beyond character animation, this work could be used to develop controllers for robotics applications that are robust to sensor noise and perturbations if the trained character model accurately reflects the robot's physical parameters.

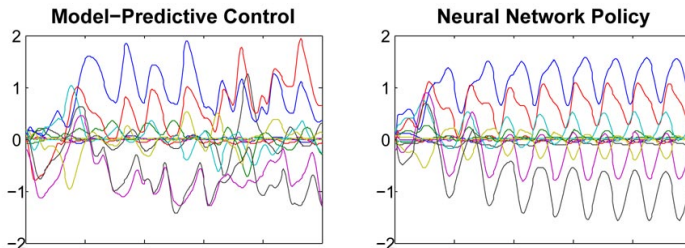

Figure 3: Typical joint angle trajectories that result from MPC and our method. While both trajectories successfully maintain position for a bird character, our method generates trajectories that are cyclic and regular.

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
