[Reviews · NeurIPS 2015]

Submitted by Assigned_Reviewer_1

The author begins by proposing a strategy to embed feedback gains in neural network policies that makes them robust to disturbances and regression error. Trajectory optimization and policy optimization are softly coupled by a cost penalty for deviation, allowing the policy to learn reasonable approximations of optimal trajectories that are in fact simpler and more robust than the products of MPC-based trajectory optimization on their own. The tasks are optimized using sophisticated parallelization, though this is not the central purpose of the paper and more of a means to an end. Finally, a diverse set of robots are controlled using the trained feedback controllers.

This paper is of the highest quality, is incredibly clear, represents a tour de force of technique, and presents quite beautiful results in its applications. Bravo.
Summary: This is a beautiful body of work that marries graphics, control, neural networks in the pursuit of generation of controllers for a diversity of body morphologies and tasks. It's inspiring.

Submitted by Assigned_Reviewer_2

"Interactive control of diverse complex characters with neural networks" shows

how a small, fast neural network can be used to represent a control policy for interactive control of a character in a physics simulator.

The approach uses trajectory optimization in a training phase to learn a neural network policy. The contribution is to show that the same architecture works across a range of more-or-less cyclic behaviours, and a range of body types.

General Comments ----------------

My reading of this paper is that it is an application of deep learning to existing (albeit recent) techniques in computer graphics.

The ability of trajectory optimization to find realistic control signals has

been established by previous work, some of which is cited here.

The use of neural networks to approximate the step-by-step actions required of a policy in order to trace out trajectories similar to the ones found by trajectory optimization has also been developed in previous work (such as e.g. [9]).

The authors of this paper must therefore work harder to position the work that went into this paper relative to these previously-published approaches.

The

closest work to my mind is the recent line of work by Pieter Abbeel and Sergey Levine. How would the authors compare the approach here to that one? Is it meant to learn faster? To support a wider variety of body shapes? Is it meant to require less computation at runtime? To be more robust to certain environmental variabilities?

Specific Comments -----------------

What is meant by "we found the resulting process can produce very large and ill-conditioned feedback gains"?

Consider moving the discussion of why you did not use LQG until after you have presented your strategy.

The notation around lines 226-240 is confusing. You introduce s(X) = \bar s, and then define a cost in terms of s(X) - \bar s. The bars and squiggles are visually similar.

What is meant by "our approach produces softer feedback gains according to parameter \lambda"? Softer than what? What does it mean for gains to be either soft or not soft? Why is it a good thing to change the initial state as part of your procedure?
Summary: Sound work, but the contribution relative to other recent related work is not clear enough.

Submitted by Assigned_Reviewer_3

In this submission, authors propose an algorithm for real-time control of 3D models, where a neural network (NN) is used to generate animations, after having been trained offline to reproduce the output of Contact-Invariant Optimization (CIO). Importantly, CIO and NN training are not independent: they are performed in an alternating fashion, the output of one being used in the criterion optimized by the other. Another core element of the method is the injection of noise, through data augmentation by small perturbations as well as additive noise in the NN's hidden layers. Experiments show that such a network is able to produce realistic and stable control policies on a range of very different character models.

This is a very interesting approach, considering that real-time control of arbitrary models is a challenging task. Although the proposed method may initially appear relatively straightforward ("learn a NN to predict the output of CIO"), the additional steps to make it work (joint training and noise injection) are significant new contributions, and shown in experiments to help get much better generalization.

Those experimental results would be stronger, however, if comparisons were made with a larger dataset. Since the base training trajectories were generated with only "between 100 to 200 trials, each with 5 branched segments", it is not surprising for the "no noise" variant to overfit. The "no joint" variant might also benefit from more diverse training trajectories -- even if that is less obvious.

Overall the comparative evaluation is the weak point of the paper (acknowledged in the introduction: "A systematic comparison of these more direct methods with the present trajectory-optimization-based methods remains to be done"). The only comparison to a competing method is a quick one with MPC at the end of the paper (and there is no reference for this method by the way).

For the most part, the maths are clearly explained and motivated, although someone willing to replicate these results will definitely need to read up additional references, since some points are only addressed superficially (especially the CIO step -- see Eq. 6 whose notations are not defined). Providing code would certainly be much appreciated by the community. There is one part I was not able to properly understand: section 3.2 about generating optimal actions for noisy inputs. Notations got really confusing for me at this point, and it was not clear to me what was being done (in particular the time index "t" was dropped, but is it still implied for some of the quantities?).

It would have also been useful to get some time measurements for computations during real-time control, since that could be a major bottleneck (and also knowing how these computations scale with the complexity of the 3D model). In an application like a video game, there is not much CPU available to animate a single character on screen.

Additional minor remarks: - "biped location": locomotion? - "It is our aim to unity these disparate approaches": unite? unify? - Am I right that the target is only about x,y,z coordinates, and there is no constraint on the angle the character is facing when reaching it? If yes, would it be easy to add an extra target angle? (which seems important in practice) - I would not call "noise injection" the step where additional data is generated from noisy inputs, because the target is re-computed to match the modified inputs. To me, this is simply "data augmentation", and it has nothing to do with the idea behind denoising autoencoders. - Are you really using the initial (random) network weights theta in the very first iteration, to compute the first X's in Alg. 1? Or are you starting with X's computed without the regularization term? - What is the justification for using the same hyperparameter eta in the two steps of Alg. 1? - "it is non-trivial to adapt them to asynchronous and stochastic setting we have": missing "the"? - Acronym LQG is used one line before it is defined. - "While we can use Linear Quadratic Gaussian (LQG) pass": missing "a"? - l. 231 the star in the argmin is in the wrong place, and is \bar{s} missing in \tilde{C}? - "the minimizer of a quadratic around optimal trajectory": missing a word? - "sX and aX are subsets of \phiX": why is that guaranteed? - "a single pass over the data to reduce the objective in (4)": it would rather be the objective from Alg. 1 since only theta is optimized - The asynchronous joint optimization scheme described in Section 6 is very different from the alternating scheme from Alg. 1. It would be worth at least mentioning this earlier. - l. 315, f should be f^{des}? - "In the current work, the dynamics constraints are enforced softly and thus may include some root forces in simulation.": I did not understand this sentence - The "no noise" variant in experiments could be split between "no data augmentation" and "no hidden noise injection". - A question I would find interesting to investigate is whether alpha can be decreased all the way to zero during optimization (for CIO only). In other words, is it working mostly because of a "curriculum learning" effect (easier task initially), or is it always useful to keep the trajectories learned by CIO close to what the network can learn? - Caption of Table 1a says 4 layers but I believe this is 3
Summary: A great-looking application of neural networks to character control, whose practical benefits and applicability still need to be established.

Submitted by Assigned_Reviewer_4

This paper proposes to combine trajectory optimization with policy learning to enable interactive control of complex characters. The main proposal is to jointly learn the trajectories and the policy, with the additional empirical insight that decoupling the two optimization problems into alternating subproblems i) enables reusing of existing, well documented methods and ii) works sufficiently well in practice. Although I could follow the paper well, I'm not immersed in this domain well enough to comment on the potential impact of this paper, which on the surface certaintly looks like a technical prowess albeit with no novel theoretical insights.

The authors combine an impressive level of technical sophistication at all levels - from the concepts down to their implementation - with equally impressive results. I found the claimed advantages of i) using sensory noise during training and ii) learning trajectories jointly with the policy, to have been clearly demonstrated.

I also found the paper was clearly written, with what I think is the appropriate level of detail given the complexity of the problem and its implementation. Each section treats a separate aspect of the problem and there are very useful cross-references between sections.

In think the motivations for the extra optimization setp in sec 7 could have been better explained. At this stage, given what had been announced in previous sections, I was somehow expecting the policy to run without the need of any extra optimization.

Is that because there is always a residual trajectory cost (or policy regression error?) such that feeding the action given by the policy into the physical simulator would accumulate errors? I would imagine that the forward evolution of the dynamics takes into account physical constraints so that no physical implausibility can result anyway, but I might be missing something? I appreciate, though, that re-optimizing at every timestep lets you use larger timestep.

l153: similar to recent approaches... can you perhaps give one or two refs? It's counterintuitive to me that keeping updates small in parameter space would be a sensible thing to do in general, though in the end I understood that it's useful in a scenario like yours where you're block-alternating between subproblems.

typo: l89: to unify
Summary: The paper's results look rather impressive, and the proposed optimization scheme + implementation sensible to me. However this research area is slightly off my radar so it's difficult for me to be confident about novelty and significance.

Submitted by Assigned_Reviewer_5

- It would be nice to mention from the beginning that you need full model knowledge (Eq. 6)

- Sect. 4 and 6 are disproportionate.

- Sect. 4 does not mention the recurrent units from the abstract.

- l. 405: How is this a stochastic NN? The training data is stochastic but the network itself seems deterministic.

Rebuttal ======== Thank you for the additional information.
Summary: LIGHT REVIEW The paper shows promising results but should be restructured.

Author Feedback
Author rebuttal: We thank the reviewers for their insightful comments. Several reviewers inquired about the comparison to related methods, in particular work of Levine and Abbeel. We address this below, but as a meta-comment, note that these papers are not based on easy-to-implement algorithms that can be compared side-by-side on established benchmarks. Hopefully this emerging field will settle at some point, and more direct comparisons will become feasible. But for now, we have to assume that each group has pushed their methods as far as they could, and obtained the best results that are possible with reasonable effort. So we can compare the end results. This of course is not as solid as a more controlled comparison, but it is the best we can offer for now.

The concurrent line of work by Levine and Abbeel in application to locomotion demonstrated neural network policies on steady non-interactive controllers for a 2D biped and a swimmer. Furthermore, their method requires an existing biped controller (Simbicon) to initialize trajectory optimization and would not generalize to non-biped morphologies. We use contact-invariant optimization to generate stable 3D locomotion, which does not require any manual initialization; we simply repeat the initial pose for initialization.

Furthermore, our algorithm is designed to be asynchronous and deployable on large compute clusters while still converging. Trial trajectory states or goals can be corrupted by noise and entire trials can be added or removed. In contrast, approach of Levine and Abbeel requires keeping track of Lagrange multipliers for every state and trial and thus must operate on a fixed set of trials.

We also easily incorporate recurrent memory into our method, and have experimented with training general RNNs with the block-alternating optimization described in the paper and found it to outperform BPTT and LSTM on initial synthetic experiments. We are happy to include these results in the supplementary material of the final submission.

Reviewers 2 and 3 questioned the necessity and performance of an extra run-time optimization step in section 7. This step is necessary because our policy commands high-level control features, such as desired joint velocity and contact forces, rather than directly commanding torques. The motivation is that this set can potentially include other more compact and high-level features, such as character's end effector positions or center of mass. As a result, we need an extra step to convert our high-level control features to control torques that would be passed to the simulator. There are many standard ways in the graphics and robotics literature to compute torques from control features, and we used the one that is simplest to implement in our pipeline (doing one-step trajectory optimization). Other ways, such as operational-space control can be implemented instead. Our implementation runs at 60 FPS on a modern desktop machine and there exist real-time implementations of operational/feature-space control in game production settings [1]. We have also experimented with commanding torques directly for bird and swimmer scenarios and found the resulting policies to work successfully without performing an extra optimization step.

Reviewer 1 asked for clarification on our feedback gain comparison. We observed that feedback gain matrices produced by LQG are an ill-conditioned function of the state, in particular producing very large spikes around contact events and resulting in unstable controllers when used without tuning. Our approach is a novel way to regularize feedback gains to be a smoother function without changing the underlying objective.

Reviewers 2 and 3 asked a number of questions, some of which we will address here.

Our target does not only include x,y,z position of the character. We are able to control the target heading of the character as well (as shown in the video for quadruped and swimmer examples).
We do use random network weights for the first algorithm iteration. Not using regularization term on first iteration does speed up convergence, but for simplicity and consistency, we do not treat first iteration specially.
A term that keeps optimized parameters close to previous iteration's solution has been used in stochastic ADMM [2]
a question of whether our regularization term is only necessary early in the optimization and can be removed at the end is interesting! Our initial experiments show that policy performance indeed degrades if this term is removed at later stages of optimization. In other words, this term actually shapes the nature of the behaviors, rather than just acting as an optimization tool.

[1] Evangelos Kokkevis, Practical Physics for Articulated Characters, Game Developer's Conference 2004.
[2] Hua Ouyang et al, Stochastic Alternating Direction Method of Multipliers, ICML 2013